# Immune-Related Thyroiditis in Patients with Advanced Lung Cancer Treated with Immune Checkpoint Inhibitors: Imaging Features and Clinical Implications

**DOI:** 10.3390/cancers15030649

**Published:** 2023-01-20

**Authors:** Hyesun Park, Akinori Hata, Hiroto Hatabu, Biagio Ricciuti, Mark Awad, Mizuki Nishino

**Affiliations:** 1Department of Radiology, Brigham and Women’s Hospital, Boston, MA 02115, USA; 2Department of Diagnostic and Interventional Radiology, Osaka University Graduate School of Medicine, Osaka 565-0871, Japan; 3Department of Medical Oncology, Dana-Farber Cancer Institute, Boston, MA 02215, USA

**Keywords:** immune checkpoint inhibitor, nonsmall cell lung cancer, immune-related thyroiditis, immune-related adverse event

## Abstract

**Simple Summary:**

An immune-related adverse event (irAE) is a unique side effect related to immunotherapy involving many different organs. Immune-related thyroiditis is one of the irAEs, and some patients experience permanent hypothyroidism requiring hormone replacement therapy. There are not many articles investigating the imaging findings of immune-related thyroiditis. We aimed to investigate the incidence and imaging characteristics of restaging chest CT scans, which are routinely performed per oncologic protocol. We found that approximately 10% of patients showed radiologically evident thyroiditis. The gland showed diffuse hypoattenuation or heterogeneous attenuation changes, with enlargement or atrophy of the gland on CT scans. In addition, among patients with thyroiditis, the patients with positive imaging findings were managed more frequently with hormone replacement therapy compared with the patients without positive imaging findings. This article provides an imaging finding of thyroiditis on restaging chest CT scans during immune checkpoint inhibitor therapy that may alert clinicians to the presence of clinically relevant thyroiditis.

**Abstract:**

Immune checkpoint inhibitors (ICI) are widely used in advanced nonsmall cell lung cancer (NSCLC) treatment, and the immune-related adverse events involving many organs have been recognized. This article investigated the incidence and imaging characteristics of immune-related thyroiditis in NSCLC patients and correlated the findings with clinical features. A total of 534 NSCLC patients treated with ICI were included. Imaging findings indicative of thyroiditis included changes in morphology and attenuation on restaging chest CT scans and FDG uptake on PET/CT during ICI therapy. Fifty patients (9.4%) had imaging findings indicative of thyroiditis. The median time to onset was 9.5 weeks (range: 0.9–87.4 weeks). The most common finding was diffuse hypoattenuation of the gland (72%), with enlargement in 15 and atrophy in 12 patients. Heterogeneous attenuation of the gland was noted in 12 patients (24%), with enlargement in 7 and atrophy in 1 patient. Two patients (4%) showed increased FDG uptake in the gland on PET/CT without changes in the CT scan. Twenty-two patients who had both clinical and radiologic diagnoses of thyroiditis were more frequently managed with hormone replacement than those with thyroiditis without an imaging abnormality (*p* < 0.0001). Therefore, awareness of the imaging findings of immune-related thyroiditis may alert clinicians to the presence of clinically relevant thyroiditis.

## 1. Introduction

Immune checkpoint inhibitor (ICI) therapy has brought another paradigm shift in advanced cancer treatment [1,2]. With the widespread use of ICIs, there has been increasing attention to the adverse events related to therapy. An immune-related adverse event (irAE) is a unique phenomenon that triggers an immunologic reaction toward normal body organs, with manifestations similar to autoimmune disease. Common irAEs include dermatitis, enterocolitis, and pneumonitis [3,4,5,6].

Immune-related thyroiditis is one of the most common endocrinologic immune-related adverse events (irAEs) [7,8,9]. The incidence of immune-related thyroiditis has been reported to be between 6–11%, and it typically presents as elevated thyroid hormone levels followed by hypothyroidism, or as isolated hypothyroidism. Most patients eventually require long-term hormone replacement therapy due to hypothyroidism [7,8,9,10].

Imaging has become an indispensable component in the evaluation of patients treated with ICIs not only for assessing the treatment response but also for identifying irAEs. Several previous articles have described the imaging findings of thyroiditis associated with immune checkpoint inhibitors including FDG-PET/CT or sonographic findings [11]; however, very few reports describe the CT imaging features [3,4]. As CT is the most commonly used imaging modality for the staging and evaluation of treatment response and adverse events in most cancer patients, a more comprehensive understanding of the CT scan findings of immune-related thyroiditis could potentially benefit patients through early detection of thyroiditis.

The purpose of this study is to investigate the incidence and imaging characteristics of radiologically evident immune-related thyroiditis and correlate them with clinical features including patient symptoms and thyroid function tests.

## 2. Material and Methods

This retrospective study was approved by the institutional review board and was Health Insurance Portability and Accountability Act (HIPPA)-compliant. Medical record and imaging reviews were retrospectively reviewed for all patients who had consented to a correlative research study approved by the institutional review board.

### 2.1. Patients

A total of 629 patients with lung cancer who were treated with ICI therapy using PD-1/PD-L1 inhibitors, either alone or in combination with CTLA-4 inhibition, at a single large tertiary care cancer center from July 2011 to October 2018 were included in our study. After excluding patients without baseline or follow-up chest CT scans, 534 patients with baseline and at least one subsequent chest CT imaging constituted the study cohort. Clinical data including patient symptoms (i.e., asymptomatic/subclinical or symptoms related to hyperthyroidism or hypothyroidism such as palpitations, tremors, heat intolerance, weight loss, and fatigue), serum thyroid-stimulating hormone (TSH), and management history (i.e., observation, beta blocker, thyroid hormone replacement) were collected from the electronic medical record. Hypothyroidism was defined by a TSH level higher than the upper limit of normal (normal institutional TSH range: 0.3–5.0 mIU/L). Hyperthyroidism was defined by a low or undetectable serum TSH level.

### 2.2. Imaging Review

Two fellowship-trained radiologists (H.P. and A.H) independently reviewed the baseline and serial scans (chest CT plus FDG-PET/CT if available) of patients who received ICI therapy. The readers were aware that the patients received ICI therapy; however, they did not have access to other clinical records. A chest CT or PET/CT scan was performed according to the treating clinical providers’ discretion without predefined intervals. Image reviews were performed up to 2 years after start of ICIs, or until the initiation of the next systemic therapy, whichever occurred first. The selection of the 2-year period was based on the results of a previous study showing that immune-related thyroiditis typically occurs within 2 years following the initiation of therapy [12]. The date of the first positive scan and the imaging findings suggestive of immune-related thyroiditis were recorded as previously described [4,13].

The imaging findings from the CT scans were recorded as the morphology (new enlargement vs. atrophy vs. no change in size) and attenuation (diffuse hypodense vs. heterogeneous) of the gland. On PET/CT, metabolic activity was categorized as (1) increased FDG uptake and (2) normal. The reviewers scored the likelihood of immune-related thyroiditis using a 6-point scale; 1, definitely not irAE; 2, probably not irAE; 3, equivocal; 4, probable irAE; 5, definite irAE; 6, nonevaluable—thyroidectomy/gland not visualized on baseline CT or PET/CT scan or underlying thyroid disease with diffuse increased FDG uptake on baseline PET/CT.

After the initial review by Reader 1 and Reader 2, the cases were classified into two groups, either positive or negative for imaging findings indicative of immune-related thyroiditis. The cases were considered positive if the assigned scores were 4 or 5 (probable or definite irAEs) by both readers or if the scores were 4 or 5 by one reader and 3 by the other reader. The cases were considered negative if the assigned scores were 1, 2, 3, or 6 by both readers. The cases with discordant scores (scores of 4 or 5 by one reader and scores of 1, 2, or 6 by the other reader) were reviewed by Reader 3 to determine if the cases were positive or negative for thyroiditis.

### 2.3. Statistical Analysis

A comparison across groups was performed using a Fisher exact test for categorical variables and a Wilcoxon Rank Sum test for continuous variables. All *p* values are based on a two-sided hypothesis. A *p* value of less than 0.05 was considered to be significant. Swimmer plots were generated using the swim plot R package (version 3.6.3) to demonstrate the temporal relationship between the imaging and laboratory findings of thyroiditis as well as ICI therapy and thyroiditis treatment.

## 3. Results

### 3.1. Clinical Characteristics of the Patients

The demographics and clinical characteristics of 534 patients (249 men and 285 women, median age of 65) are shown in Table 1. The majority of patients (503/534, 94.2%) received PD-1 or PD-L1 inhibitor monotherapy, and the remainder (31/534, 5.8%) were treated with combination therapy of CTLA-4 and PD-1 inhibition. All patients had at least one follow-up chest CT scan in addition to the baseline chest CT scan. There were 45 patients (8.4%) who had a FDG-PET/CT scan in addition to the CT scan during ICI therapy among the 534 patients.

### 3.2. Immune-Related Thyroiditis Based on Imaging Findings

There were 50 patients (50/534, 9.4%) who showed imaging findings indicative of thyroiditis during ICI therapy (Figure 1). The median time from the initiation of therapy to the first CT scan with positive imaging findings was 9.5 weeks (range: 0.9–87.4 weeks). The most common CT finding for thyroiditis was diffuse hypodense attenuation of the thyroid gland, which was seen in 36 patients (36/50, 72%); 15 of these patients also demonstrated enlargement of the thyroid gland and 12 patients demonstrated atrophy (Figure 1). The heterogeneous attenuation of the gland was noted in 12 patients (12/50, 24%), with accompanying enlargement in 7 patients and atrophy in 1 patient (Figure 2). Among the 12 patients with heterogeneous enlargement, 3 patients also had a diffuse increased FDG uptake on PET/CT. Two patients (2/50, 4%) did not show any morphologic changes with the CT scan but had a diffuse increased FDG uptake of the thyroid gland on PET/CT (Figure 3). There were no significant differences in the demographic and clinical parameters between patients with and without findings suggestive of thyroiditis (Table 1)

### 3.3. Immune-Related Thyroiditis Based on Abnormal Serum TSH

There were 77 patients (55/534, 14.4%) that were diagnosed with immune-related thyroiditis during ICI therapy based on TSH levels, including 47 with hypothyroidism and 30 with hyperthyroidism at the time of the clinical presentation of thyroiditis. There were no significant differences in terms of the clinical characteristics between the patients with and without clinical immune-related thyroiditis, including drug regimens (PD-1/PD-L1 monotherapy vs. CTLA-4 and PD-1/PD-L1 combination therapy). Among these 77 patients, 40 (52%) required treatment for thyroid dysfunction, 25 developed hypothyroidism (24 needed permanent thyroid hormone supplementation and 1 recovered normal thyroid function after transient thyroid hormone supplementation), 13 developed an initial decrease in TSH with subsequent hypothyroidism (all 13 required permanent thyroid hormone replacement and 1 patient was also treated with nonsteroidal anti-inflammatory drugs for neck pain, and 2 patients presented with hyperthyroidism, 1 of which was treated with methimazole (antithyroid agent), and the other was managed with propranolol and steroid), and the remaining 37 patients (48%) were on close observation with follow-up TSH without any treatment.

Among the 77 patients with clinically diagnosed thyroiditis, ICI therapy was discontinued in eight patients; five patients stopped therapy due to irAEs other than thyroiditis (two pneumonitis, one arthritis, one colitis, and one anasarca), two patients stopped due to disease progression, and one patient had renal insufficiency and did not tolerate ICI.

### 3.4. Clinically and Radiologically Evident Immune-Related Thyroiditis

There were 22 patients who had both an abnormal serum TSH level and imaging findings indicative of thyroiditis: 13 with hypothyroidism and 9 with transient hyperthyroidism with subsequent hypothyroidism. There was no statistical difference in the imaging findings between the patients with hypothyroidism and transient hyperthyroidism with subsequent hypothyroidism. Twenty patients (91%) were treated with levothyroxine for thyroiditis and needed ongoing thyroid hormone supplementation. Among the patients who had clinically diagnosed immune-related thyroiditis, the 22 patients who also had positive CT findings of thyroiditis were treated with hormone replacement therapy more frequently than patients with clinical diagnoses alone without positive CT findings (20/22 versus 20/55; Fisher *p* < 0.0001). Among the 22 patients with both clinical and imaging findings, 7 patients had coexisting endocrinopathy, 5 had hypophysitis, 1 patient has autoimmune diabetes, and 1 patient has both hypophysitis and autoimmune diabetes. There was no severe case of thyroiditis that needed hospitalization for management or was related to death.

### 3.5. Temporal Changes in the CT Findings Correlating with TSH

In 22 patients who had clinical and radiographic thyroiditis, the median interval between the date of the first positive CT scan and the date of the first abnormal TSH level was 6 weeks (range: −75.0 to +75.4 weeks). Four patients had positive CT imaging findings before the TSH abnormality, two patients presented with thyroiditis at the same time as the positive imaging studies, and sixteen patients showed abnormal TSH levels before demonstrating morphologic changes with the CT scan (Figure 4).

Among 22 patients, 11 patients showed an unchanged size or enlargement of the gland with heterogeneous or diffuse hypoattenuation of the gland on the initial positive scan, and 10 patients eventually showed atrophy of the gland with heterogeneous or diffuse hypoattenuation of the gland on follow-up CT scans (mean: 5.8 weeks; range: 1.4 to 14.4 weeks); additionally, one patient had no follow-up imaging. Eleven patients with atrophy of the gland with heterogeneous or diffuse hypoattenuation of the gland on the initial positive CT scan demonstrated progressive atrophic changes in follow-up studies (Figure 5).

## 4. Discussion

Our study demonstrated that the incidence of radiologically evident immune-related thyroiditis was 9.4%. The diffuse hypoattenuation of the thyroid gland with the enlargement or atrophy of the gland was the most common CT finding, followed by the heterogeneous attenuation of the gland. The incidence of clinically evident thyroiditis was 14.4% with 61% of the patients presenting with hypothyroidism and 39% presenting with transient thyrotoxicosis followed by hypothyroidism. Among the patients who had clinically diagnosed immune-related thyroiditis, patients who also had positive CT findings of thyroiditis were treated with hormone replacement therapy more frequently than patients without positive CT findings.

Immune-related thyroiditis is one of the common immune-related endocrinopathies. The incidence of all-graded immune-related thyroiditis is reported to be from 5% to 9.5% with pembrolizumab treatment and 8.6% with nivolumab treatment based on clinical trial results [14,15,16]; however, a meta-analysis reported that the incidence of thyroiditis related to pembrolizumab up to 19.8% [17]. According to the meta-analysis by Barroso-Sousa et al., patients with combination ICI therapy (i.e., CTLA-4 and PD-1/PDL-1 inhibitors) were more likely to develop immune-related thyroiditis, and patients who were treated with PD-1 inhibitors had a higher risk of developing hypothyroidism compared to the patients using a CTLA-4 inhibitor, and they also had a higher risk of developing hyperthyroidism compared to the patients on PD-L1 inhibitors [7]. Our study did not show a significant difference among the patient groups who were treated with combination or monotherapy as well as among the groups who were treated with PD-1 or PD-L1 inhibitors. This may be due to the small number of patients who received combination therapy and the fact that the majority of patients received PD-1 or PD-L1 inhibitor monotherapy (250 patients with pembrolizumab and 221 patients with nivolumab).

Imaging findings of immune-related thyroiditis have been described in previous studies, mainly focusing on the sonographic or PET/CT findings. On ultrasounds, the gland shows mild enlargement with diffuse low echogenicity [18], which is similar to other thyroiditis such as Hashimoto thyroiditis. The main difference between immune-related thyroiditis and Hashimoto thyroiditis is that the gland eventually becomes atrophic in immune-related thyroiditis, whereas Hashimoto thyroiditis remains enlarged mainly with diffuse fibrotic change [19]. On PET/CT, there is transient diffuse FDG uptake of the thyroid gland reflecting active inflammation [11,20]. A positive PET/CT finding is often present before the development of clinically overt thyroiditis, which can be used as a predictive indicator [20]. This study also reported the incidence of PET-positive thyroiditis as 22%, and there was a transient increase in FDG uptake with an SUVmax of 116% (IQR: 84–177; range: 52–300); additionally, a follow-up study found a resolution of FDG uptake with a median of 32 days (IQR: 79–194; range: 49–1045) [20]. In our study, there were five patients who had positive PET/CT findings among the forty-five patients who had PET/CT during ICI, whereby three of them showed both morphologic changes and increased metabolic activity and two of them showed FDG uptake of the gland without morphologic change. The evaluation of incidence or temporal changes in positive FDG-PET/CT among the study cohort was limited, as PET/CT is not routinely performed during follow ups on lung cancer.

The CT findings of immune-related thyroiditis were described in a very few number of studies [4,13], and none have reported on CT findings in terms of the temporal morphologic changes in the CT scan. The diffuse hypoattenuation of the thyroid gland on CT scan images is well known to be correlated with abnormal thyroid function [21,22]. Our study also showed that diffuse hypoattenuation was the most common finding, which is suggestive of immune-related thyroiditis in both hypothyroidism and hyperthyroidism. The pathophysiology of immune-related thyroiditis is not well understood; however, an observational study showed that thyroid dysfunction is mainly due to silent, destructive thyroiditis [10]. The cytopathology of immune-related thyroiditis demonstrates unique findings that include abundant clusters of necrotic cells and rare typical thyroid follicular cells as well as aggregates of lymphocytes and histiocytes [23]. The abundant presence of necrotic cells may suggest a destructive mechanism of thyroiditis [23]. Another recent study showed that immune-related thyroiditis is a T-cell-mediated inflammatory process with the abundant intrathyroidal infiltration of CD8+ and CD4−CD8− T lymphocytes [24]. In this context, the initial enlargement of the gland may reflect an acute inflammatory process with inflammatory cellular infiltrates, and the progressive atrophy of the gland may represent a burnout gland after the destruction of parenchyma. Iodine is a major component of the thyroid hormone, and its incorporation into thyroid gland follicles makes the thyroid gland have an intrinsic high attenuation on CT scans. In immune-related thyroiditis, the gland appears heterogeneous or demonstrates diffuse hypoattenuation probably due to inflammatory cell infiltrates and necrotic cells [23].

The clinical manifestation of the immune-related thyroiditis ranges from asymptomatic to overt hypothyroidism, which causes fatigue and weight gain [12]. According to a previous study, 67% of patients were asymptomatic during the thyrotoxicosis phase [25]. Patients with immune-related thyroiditis needed long-term hormone replacement therapy, and a minority of patients showed clinically reversible thyroiditis without the need for long-term hormonal replacement in previous studies [25], which is similar to our study’s findings. This study also showed that the patients who had positive imaging findings suggestive of immune-related thyroiditis were more frequently treated with medication, mainly with hormone replacement therapy for hypothyroidism. The positive imaging findings may indicate the destruction of the gland with the eventual loss of function, which can be irreversible.

There are similarities in the clinical and imaging features between immune-related thyroiditis and subacute thyroiditis. Subacute thyroiditis is thought to be an immune-related thyroiditis of unclear etiology, which occurs a few weeks after a viral infection of the upper respiratory tract, which results in neck pain and swelling and possible thyrotoxicosis followed by hypothyroidism [26]. Recently, subacute thyroiditis has also been recognized as part of COVID-19 infection in the current worldwide pandemic [27,28]. On ultrasounds, subacute thyroiditis shows focal hypoechoic lesions on the thyroid gland, which mimics thyroid cancer. The long-term clinical course between the two entities is different in that subacute thyroiditis usually recovers and permanent hypothyroidism is uncommon, whereas the majority of immune-related thyroiditis develops irreversible glandular destruction and progressive atrophy.

There are several limitations to our study. First, this is a retrospective design study whereby we reviewed images and the medical records of patients treated in a single institution. Given the low incidence of immune-related thyroiditis, it is important to validate the results with a larger study cohort in a multicenter study in the future. Second, the TSH results were not available at the time of the positive scan in some patients, which made it difficult to evaluate the relationship between the timing of CT positivity and abnormal TSH. Lastly, there was no pathologic correlation for the imaging findings because no patient underwent a thyroid gland biopsy or surgery. However, we correlated them with abnormal TSH values, which is a reference standard for thyroiditis.

## 5. Conclusions

In conclusion, radiologically evident immune-related thyroiditis during ICI therapy was seen in 9.4% of advanced NSCLC patients. The most common CT finding of immune-related thyroiditis was the diffuse hypoattenuation of the gland, with transient enlargement or atrophy. Among clinically evident immune-related thyroiditis, the patients with positive imaging findings were more frequently treated with hormone replacement therapy. Awareness of the imaging findings of immune-related thyroiditis on chest CT scans is important because of the therapeutic implications for thyroiditis.

## Figures and Tables

**Figure 1 cancers-15-00649-f001:**
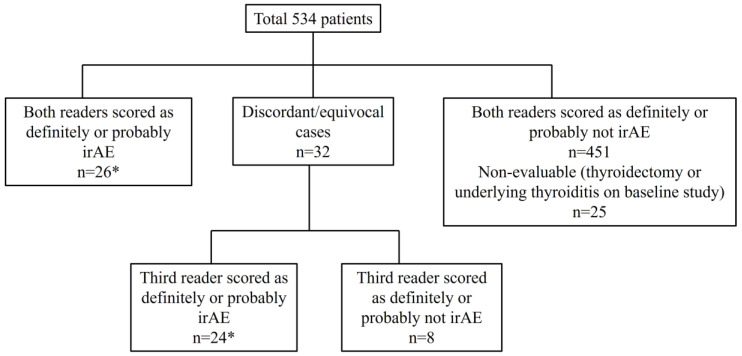
Flow chart for the radiologic review of readers. A total of 50 patients (marked as asterisks) showed CT findings indicative of immune-related thyroiditis.

**Figure 2 cancers-15-00649-f002:**
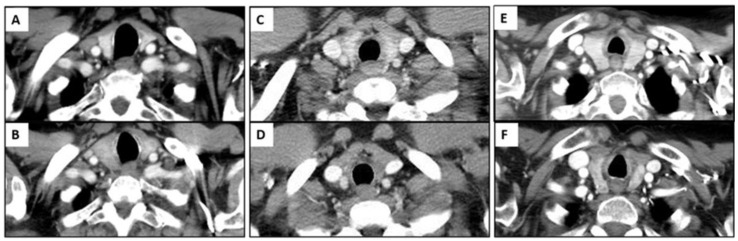
Imaging findings of immune-related thyroiditis from chest CT scan. (**A**,**B**) A 75-year-old man with advanced lung cancer treated with nivolumab monotherapy. There was interval development of enlarged thyroid gland with diffuse hypoattenuation after 4 weeks of therapy (**B**) compared to the baseline image (**A**). (**C**,**D**) A 79-year-old woman with advanced lung cancer treated with pembrolizumab monotherapy. There was interval significant atrophy of thyroid gland with diffuse hypoattenuation after 7.1 weeks of immunotherapy (**D**) compared to the baseline image (**C**). (**E**,**F**) A 78-year-old woman with advanced lung cancer treated with nivolumab monotherapy. There was interval atrophic change in thyroid gland with heterogeneous attenuation after 13.4 weeks of therapy (**F**) compared to the baseline image (**E**).

**Figure 3 cancers-15-00649-f003:**
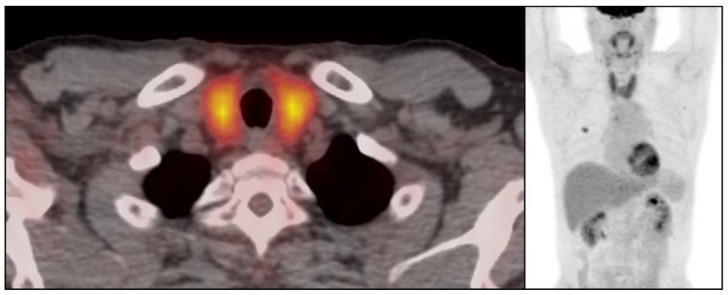
Imaging findings of immune-related thyroiditis on FDG-PET/CT. 57-year-old woman with advanced lung cancer treated with nivolumab monotherapy. After 7.9 weeks of therapy, there was a diffuse FDG uptake of thyroid gland representing immune-related thyroiditis.

**Figure 4 cancers-15-00649-f004:**
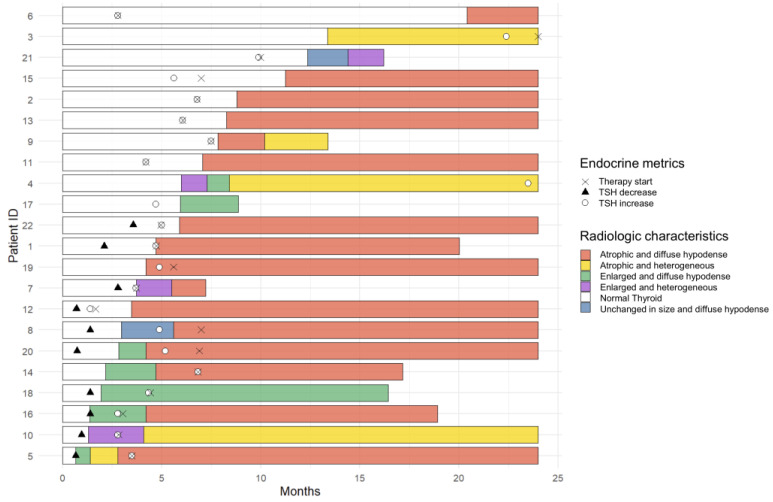
Temporal changes in the CT findings correlating with TSH level and the start of hormone replacement therapy in each patient among 22 patients who had both clinical and radiologic immune-related thyroiditis. Four patients showed imaging abnormality before TSH abnormality, and all demonstrated hypothyroidism. The rest of 18 patients demonstrated abnormal TSH levels at the same time or before the imaging abnormality (10 thyrotoxicosis, 8 hypothyroidism). Black triangle indicates the time point when the patient had thyrotoxicosis (TSH < 0.3 mIU/L) and white circle indicates the time point when the patient had hypothyroidism (serum TSH > 5.0 mIU/L). Cross indicates the date of starting thyroid hormone replacement therapy.

**Figure 5 cancers-15-00649-f005:**
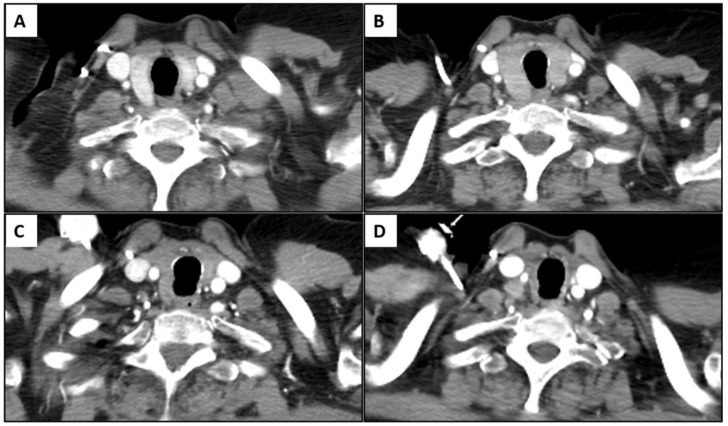
Representative case of temporal change in immune-related thyroiditis. A 70-year-old female with advanced lung cancer treated with ipilimumab and pembrolizumab combination therapy. There was interval enlargement of thyroid gland with diffuse hypoattenuation (**B**) compared to the baseline study (**A**) after 3.9 weeks of therapy. On follow-up scan after 12 weeks of therapy (**C**), the gland became atrophic and progressed in serial follow ups. End-of-therapy scan (after 2 years of therapy) showed significant atrophy of gland (**D**). The patient experienced transient thyrotoxicosis followed by hypothyroidism which required hormone replacement.

**Table 1 cancers-15-00649-t001:** Demographic and clinical parameters.

		CT Positive for Thyroiditis (n = 50)	CT Negative for Thyroiditis (n = 484)	All Patients (n = 534)	*p* Value
**Age**	**Median (years)** **(range)**	63 (39–91)	66 (25–92)	65 (25–92)	0.22
**Sex**	**Male**	20	229	249	0.37
	**Female**	30	255	285	
**Smoking**	**Never**	3	73	76	0.089
	**Current/Former**	47	411	458	
**Pathology**	**Adenocarcinoma**	39	373	412	1.00
	**Other**	11	111	122	
**ECOG PS**	**0-1**	40	400	440	0.69
	**≥2**	10	84	94	
**ICI regimen**	**PD-1/PD-L1 monotherapy**	46	457	503	0.51
	**PD-1/CTLA-4 combination therapy**	4	27	31	
**Line of therapy**	**1st line**	13	141	154	0.74
	**≥2nd line**	37	343	380	

## Data Availability

The datasets generated or analyzed during the study are available from the corresponding author upon reasonable request.

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
