# Peer review of "Immune-Related Thyroiditis in Patients with Advanced Lung Cancer Treated with Immune Checkpoint Inhibitors: Imaging Features and Clinical Implications"

_cancers, 2023, doi:10.3390/cancers15030649_

Round 1

Reviewer 1 Report

In the current study the authors aim to investigate the incidence and imaging characteristics of radiologically evident immune-related thyroiditis and correlate with clinical features including patient symptoms and thyroid function tests. The study showed that diffuse hypoattenuation was the most common findings suggestive of immune-related thyroiditis in both hypothyroidism and hyperthyroidism.

The study has the potential to have scientific and translational impact, pending a series of revisions, as follows:

- “Results 3.4. Clinically and radiologically evident immune-related thyroiditis: Among the patients who had clinically diagnosed immune-related thyroiditis, the 22 patients who also had positive CT findings of thyroiditis were treated with hormone replacement therapy more frequently than patients with clinical diagnoses alone without positive CT findings (20/22 177 versus 20/55; Fisher p<0.0001)”.  The description of cycle and time for the drug use was not specific. Whether these patients have clinical manifestations should be described and displayed.

- “Results 3.5. Four patients had positive CT imaging findings before the TSH abnormality, two patients presented as thyroiditis at the same time of positive imaging studies, and 16 patients showed abnormal TSH before morphologic changes on CT scan (Figure 3). “. The description of Figure 3 in the results does not exactly match the picture shown.

- Since this is a retrospective design study with review of images in a single institution and the sample is relatively small for patients with positive CT finding. Larger sample from more institutions should be collected to verify the main results in this paper.

- The relationship between timing of CT positivity and abnormal TSH were absent.

- Lastly, there was no pathologic correlation for the imaging findings. 

Author Response

Reviewer 1

In the current study the authors aim to investigate the incidence and imaging characteristics of radiologically evident immune-related thyroiditis and correlate with clinical features including patient symptoms and thyroid function tests. The study showed that diffuse hypoattenuation was the most common findings suggestive of immune-related thyroiditis in both hypothyroidism and hyperthyroidism. The study has the potential to have scientific and translational impact, pending a series of revisions, as follows:

1-1. “Results 3.4. Clinically and radiologically evident immune-related thyroiditis: Among the patients who had clinically diagnosed immune-related thyroiditis, the 22 patients who also had positive CT findings of thyroiditis were treated with hormone replacement therapy more frequently than patients with clinical diagnoses alone without positive CT findings (20/22 177 versus 20/55; Fisher p<0.0001)”.  The description of cycle and time for the drug use was not specific. Whether these patients have clinical manifestations should be described and displayed.

Response: We appreciate your comment. This study includes patients with advanced lung cancer who were treated with immune checkpoint therapy using PD-1/PD-L1 inhibitors (with small subset of patients treated with combination with CTLA-4 inhibition). The cycle and time for the drug was not included in this study, because there is lack of evidence that immune-related adverse event is dependent on dose/exposure of therapy. The patients who were managed with hormone replacement therapy showed symptoms probably related to abnormal thyroid function (fatigue, loss of energy), whereas the patients who were asymptomatic were followed up closely with serial TSH without any treatment. The decision to manage the patient with either hormone replacement therapy or observation was made by oncologists based on the patient’s clinical presentation.

1-2. “Results 3.5. Four patients had positive CT imaging findings before the TSH abnormality, two patients presented as thyroiditis at the same time of positive imaging studies, and 16 patients showed abnormal TSH before morphologic changes on CT scan (Figure 3). “. The description of Figure 3 in the results does not exactly match the picture shown.

Response: We apologize for the error in figure labels. It is revised correctly.

1-3. Since this is a retrospective design study with review of images in a single institution and the sample is relatively small for patients with positive CT finding. Larger sample from more institutions should be collected to verify the main results in this paper.

Response: We appreciate the comment. It is a limitation of our study that the study was performed retrospectively in a single institution. We included the patients from 2011 to 2018 who were treated in the one of the large cancer centers. The purpose of this study was to investigate the incidence and CT imaging findings of immune-related thyroiditis, which has not been described previously. We agree that given the low incidence of immune-related thyroiditis, it is important to validate the results with a larger number of study cohort in the multicenter study in the future. 

1-4. The relationship between timing of CT positivity and abnormal TSH were absent.

Response: Among the 22 patients with radiologically and clinically evident thyroiditis, four patients showed imaging abnormality before TSH abnormality, all of which manifested with hypothyroidism. The rest of 18 patients demonstrated abnormal TSH at the same time or before the imaging abnormality, ten patients with thyrotoxicosis, and eight patients with hypothyroidism. The detailed temporal changes the imaging and thyroid function test result are depicted on figure 4.

1-5. Lastly, there was no pathologic correlation for the imaging findings. 

Response: We appreciate the comment. There was no pathologic correlation for the imaging findings because no patient under-went thyroid gland biopsy or surgery, which was included in the limitation of this study. However, we correlate with abnormal TSH values, which was a reference standard for thyroiditis.

Reviewer 2 Report

Authors demonstrated the characteristics of immune-related thyroiditis, regarding imaging features and clinical implications. I would like to make some comments on this manuscript.

1. Page 2, line 79, in 2.2. imaging review section

Who is two fellowship-trained radiologist? Are they also co-authors? If so, authors should clarify this. 

2. Result part

In this study, 50 patients are diagnosed as thyroiditis from CT. 77 patients were diagnosed as thyroiditis from TSH. 22 patients had both abnormal TSH and imaging findings. I do not understand who is really thyroiditis and requires treatment. From this manuscript, I'm not sure who decides thyroiditis and how to diagnose the definite thyroiditis. Are endocrinologists one of the co-authors of this manuscript? 

3. Line 323 

Authors declared this study was approved institutional review board. If so, they should clarify the approval number and approval date.

Author Response

Reviewer 2

Authors demonstrated the characteristics of immune-related thyroiditis, regarding imaging features and clinical implications. I would like to make some comments on this manuscript.

2-1. Page 2, line 79, in 2.2. imaging review section. Who is two fellowship-trained radiologist? Are they also co-authors? If so, authors should clarify this. 

Response: Two radiologists are both co-authors (H.P and A.H). We included this to the manuscript.

2-2. Result part

In this study, 50 patients are diagnosed as thyroiditis from CT. 77 patients were diagnosed as thyroiditis from TSH. 22 patients had both abnormal TSH and imaging findings. I do not understand who is really thyroiditis and requires treatment. From this manuscript, I'm not sure who decides thyroiditis and how to diagnose the definite thyroiditis. Are endocrinologists one of the co-authors of this manuscript? 

Response: We appreciate the comment. We defined thyroiditis in two methods; one is “radiologically evident thyroiditis” which is defined as the imaging characteristics of thyroiditis as described on previous studies (reference 4 and 13), and the other is “clinically evident thyroiditis” which is defined as abnormal thyroid function test. Hypothyroidism was defined by a TSH level higher than the upper limit of normal (normal institutional TSH range, 0.3–5.0 mIU/L). Hyperthyroidism was defined by a low or undetectable serum TSH level). Diagnosis and management of thyroiditis was based on the symptom and thyroid function test which were performed per oncologic follow up protocol for the patients who received immunotherapy. All clinical information including thyroid function test was retrospectively collected from medical chart review.  Two oncologists (B.R. and M.A) contributed to our study. In our study, the group of patients who had both abnormal function and imaging characteristic of thyroiditis were more commonly managed with hormone replacement therapy compared to the patients with abnormal thyroid function without imaging abnormality, which may suggest that the positive radiologic finding (i.e., morphologic and attenuation change of thyroid gland) may indicate clinically significant thyroiditis.

Response:

2-3. Line 323 

Authors declared this study was approved institutional review board. If so, they should clarify the approval number and approval date

Response: The study was performed under the IRB protocol DFHCC#02-180 which has been active since the original approval date of 9/11/2002. We included this to the manuscript.

Reviewer 3 Report

The article is without obvious deficits and I have no comments to offer to further improve. Recommend Accepting in current form.

Author Response

Thank you so much for the review.

Round 2

Reviewer 2 Report

Thank you very much for the authors' reply.

I would like to make additional comments.

1. I understand the definition of the diagnosis and management of thyroiditis. I would recommend the authors to clarify this point in the method part of the manuscript.

2. Besides, I have a question. What are the advantages of the combination of abnormal thyriod function and imaging characteristics? As far as I understand from this manuscript, among the 22 cases with CT findings and abnormal TSH, 11 had known CT findings and TSH changes BEFORE treatment (Patient ID #5,10,16,18,14,20,8,7,19,1,3). In other words, the positive predictive value (PPV) was 50%. On the other hand, of the 77 patients with TSH changes, 40 required treatment. The PPV was 52%. It is suggested from this result that the additional parameter of CT change has not led to an improvement in PPV. 

3. I have another comment on Line 75 "a single, large, tertiary care center". I don't understand why it has to be anonymous. I believe it is important to clearly indicate which data was used.

Author Response

1. I understand the definition of the diagnosis and management of thyroiditis. I would recommend the authors to clarify this point in the method part of the manuscript.

Response: Thank you for the suggestion. It is included in the method section of the manuscript.

2. Besides, I have a question. What are the advantages of the combination of abnormal thyroid function and imaging characteristics? As far as I understand from this manuscript, among the 22 cases with CT findings and abnormal TSH, 11 had known CT findings and TSH changes BEFORE treatment (Patient ID #5,10,16,18,14,20,8,7,19,1,3). In other words, the positive predictive value (PPV) was 50%. On the other hand, of the 77 patients with TSH changes, 40 required treatments. The PPV was 52%. It is suggested from this result that the additional parameter of CT change has not led to an improvement in PPV.

Response: Thank you for your comment. We noticed that among the patients who had clinically diagnosed immune-related thyroiditis, the patients who also had radiographic evidence of thyroiditis were more frequently treated with hormone replacement compared to patients without radiographic changes of gland on CT (20/22 versus 20/55; Fisher p<0.0001). Based on this result, we hypothesized that the patients with both radiographic and clinical evidence of thyroiditis might have more severe form of thyroiditis which required management with hormone. According to the previous literature, the cytopathology of immune-related thyroiditis showed abundant clusters of necrotic cells and rare typical thyroid follicular cells as well as aggregates of lymphocytes and histiocytes, which leads to destruction of gland. Therefore, morphologic and attenuation changes on CT may reflect the permanent damage of the thyroid gland.

3. I have another comment on Line 75 "a single, large, tertiary care center". I don't understand why it has to be anonymous. I believe it is important to clearly indicate which data was used.

Response: It was recommended to be anonymous for review to minimize any bias. We revised as “a single large tertiary cancer center” to be clearer.

Round 3

Reviewer 2 Report

No comments at all.